# Cane Vinasses Contain Bioactive Concentrations of Auxin and Abscisic Acid in Their Composition

**DOI:** 10.3390/ijms23179976

**Published:** 2022-09-01

**Authors:** Angel Mª Zamarreño, Giancarlo Valduga, Jose Mª Garcia-Mina

**Affiliations:** 1Department of Environmental Biology, Faculty of Sciences, BIOMA Institute, University of Navarra, 31008 Pamplona, Spain; 2Department of Marketing, Timac Agro, Porto Alegre 90480-001, Brazil

**Keywords:** vinasse, phytoregulators, indoleacetic acid, abscisic acid, cytokinins, crop yield

## Abstract

Currently, high doses of vinasse are employed for the fertigation of sugarcane with positive results on yield. Usually, this effect is related to the presence of mineral nutrients in its composition as well as to its action on soil properties. Consequently, the concentrations of minerals, organic acids, and other metabolites in vinasse are very well characterized. However, considering that cane vinasses are obtained from the treatment of vegetal tissues, it is also possible that they might contain significant concentrations of phytoregulators that could have a relevant role in their beneficial action on yield. To investigate this hypothesis, we analyzed the main plant hormones in 22 samples of vinasse collected in different production sites of Brazil using HPLC–mass spectrometry. The results show that both ABA and IAA present concentrations in vinasse within the micromolar range, thus being potential active ingredients affecting plant development. In conclusion, the beneficial action of cane vinasses on sugarcane yield might involve, among other factors, the action of IAA and ABA on plant growth.

## 1. Introduction

Sugar and alcohol production has great economic and social importance in those countries with a high productivity of sugarcane [1]. However, this industrial activity is associated with the production of by-products that can significantly impact the environment [1]. Among these by-products, vinasses have a particular relevance [1]. We can differentiate two main types of cane vinasses (vinasses throughout the article): molasses vinasse and juice vinasse, depending on whether alcohol is made from molasses or from the juice produced from cane milling. Mixed vinasse is a mixture of molasses vinasse and juice vinasse. It has been calculated that Brazil produces around 300 billion liters of cane vinasses annually [1]. In this context, developing strategies for recycling and reusing this by-product becomes very important.

One of the primary uses of vinasses is the fertigation of sugarcane [1]. It has been calculated that, on average, the dose of vinasses in fertigation is around 306 m^3^ per ha and cycle [1]. Considering the high dose applied, vinasses have been extensively characterized in their content of mineral nutrients and heavy metals [1]. Likewise, the contents of organic compounds such as organic acids, alcohols, polyphenols, and amino acids have been well characterized [1]. These data have allowed a precise evaluation of the potential risks and nutritional benefits derived from these vinasse components when used in fertigation [1].

However, a family of plant metabolites can have a high impact on plant growth and soil microbiota that has not been characterized yet, and that can be present in vinasses at significant concentrations. We refer to the concentration of the main phytoregulators. We hypothesize that as these compounds are present in sugarcane tissues, they can be concentrated in vinasses while producing sugar or ethanol. Consequently, the final concentration in vinasse of some of these phytoregulators might reach bioactive values. In order to investigate this hypothesis, we analyzed the concentration of the main auxins (indoleacetic acid, IAA; indolebutyric acid, IBA), cytokinins (CKs), gibberellins (GBs) and abscisic acid (ABA) in 22 samples of vinasses, including juice vinasse (8 samples) and mixed vinasse (14 samples), obtained from different production sites from Brazil. The results’ significance is evaluated considering the average dose of vinasse used in fertigation.

## 2. Results and Discussion

In Table 1, we show the results concerning those phytoregulators that presented significant concentrations in all or some of the vinasses analyzed.

All samples presented significant concentrations of IAA and ABA, although the variability between samples was very high (Table 1, Figure 1A). On average, the concentration of IAA was 396.31 ± 502 pmol g^−1^ in juice vinasse and 464.17 ± 515 pmol g^−1^ in mixed vinasse. In contrast, the average concentration of ABA was 167.34 ± 157 pmol g^−1^ in juice vinasse and 65.95 ± 75.72 pmol g^−1^ in mixed vinasse (Figure 1A). Some of the samples also presented low concentrations of some CKs, principally Z, iP and iPR riboside (Table 1). There were not found with measurable concentrations of GBs.

The variability in the obtained results is in the same range of that found for other vinasse organic components such as organic acids or amino acids as well as some mineral nutrients as N or K [1]. This variability has principally been associated with the singular features of the sugar or alcohol production procedures employed in each factory [1].

The PCA analysis of data showed that juice vinasse and mixed vinasse presented some differences principally due to the different concentration of ABA that is higher in juice vinasse (Figure 1B).

Let us consider a dose of vinasse application in fertigation of 306 m^3^ per ha and cycle [1]. We can calculate the concentration of IAA and ABA applied per m^2^ of soil surface. On average, this dose of vinasse involves the application of 0.0143 mmol for juice vinasse, and 0.0167 mmol for mixed vinasse, of IAA per m^2^; and 0.006 mmol for juice vinasse and 0.0024 mmol for mixed vinasse of ABA.

On the other hand, we can also calculate the active concentration of the two phytoregulators in the product that is in contact with plant roots. As the analytical results are expressed in pmol per g, the concentration of IAA that we have in the applied product, considering that vinasse density is around 1, is on average 0.396 µM in juice vinasse and 0.464 µM in mixed vinasse. In contrast, for ABA, this concentration is 0.167 µM in juice vinasse and 0.066 µM in mixed vinasse.

Taking into account that the range of bioactivity of these phytoregulators covers from nmol to mmol [2,3], it becomes clear that the dose applied of both IAA and ABA might play a role in the final effect of vinasse application on sugarcane productivity. These effects can be associated with root growth and functionality in the case of IAA, and with the resistance to abiotic stress in the case of ABA.

Therefore, the results presented here indicate that besides the nutritional effects derived from the presence of mineral nutrients, the significant concentration of two relevant phytoregulators, IAA and ABA, might also be involved in the mechanism responsible for the positive action of vinasses on sugarcane yields.

On the other hand, it is known that both IAA and ABA can produce deleterious effects on plant growth at high concentrations [2,3]. This fact suggests the convenience of controlling the concentration of these phytoregulators in vinasse before its use for the fertigation of sugarcane production.

## 3. Materials and Methods

Vinasses were collected during alcohol production and stored in coated plastic tanks protected from light. The air was displaced with the liquid to avoid oxidation. Each sample of five liters corresponded to the mixture of individual samples collected over one day. Samples were collected from factories located in diverse areas of Brazil.

The samples, without any further treatment, were stored at 15 °C before analysis. The tanks with samples were opened just before analysis. Previous studies showed that the stability of the analyzed hormones in vinasses in the absence of light and air is very high.

The following plant hormones were studied: cytokinins (CKs), zeatin (Z), dihydrozeatin (DHZ), *trans*- and *cis*-zeatin riboside (t-ZR and c-ZR), dihydrozeatin riboside (DHZR), isopentenyladenine (iP), isopentenyladenosine (iPR), the topolin family, indole-3-acetic acid (IAA), indolebutiric acid (IBA), abscisic acid (ABA), and gibberellins 1, 3, 4, and 7 (GA1, GA3, GA4, GA7). The extraction and purification of the different plant regulators were carried out directly in the samples using the method described by Dobrev and Kaminek [4] and Aguirre et al. [5].

Liquid chromatography–mass spectrometry quantification of the different phytoregulators was made by HPLC linked to a 3200QTRAPLC/MS/MS system (Applied Biosystems/MDS Sciex, Markham, ON, Canada), equipped with an electrospray interface.

The methodology used was described in Mora et al. [6].

For CKs, detection and quantification were performed by multiple reaction monitoring (MRM) in the positive-ion mode, employing a multilevel calibration graph with deuterated CKs as internal standards. Compound-dependent parameters were described by Aguirre et al. [5]. The source parameters were: curtain gas: 25.0 psi, GS1: 50.0 psi, GS2: 60.0 psi, ion spray voltage: 5000 V, CAD gas: medium, and temperature: 600 °C.

For the other hormones, detection and quantification were performed by MRM in the negative-ion mode, employing a multilevel calibration graph with deuterated hormones as internal standards. Compound-dependent parameters were described by Aguirre et al. [5]. The source parameters are: curtain gas: 25.0 psi, GS1: 50.0 psi, GS2: 60.0 psi, ion spray voltage: 4000 V, CAD gas: medium, and temperature: 600 °C.

Principal component analysis (PCA) was made by using PAST 4.09 software (Øyvind Hammer—https://www.nhm.uio.no/english/research/infrastructure/past/—Norway).

## 4. Conclusions

In summary, the study presented here demonstrates that vinasses are significant carriers of two main types of plant hormones: the auxin IAA and ABA. In fact, their concentrations, µM in average, in raw vinasse are within the bioactive range. Therefore, it is plausible that the mechanisms responsible for the beneficial action of sugarcane fertigation with vinasse might involve a direct biostimulant action of these phytoregulators.

On the other hand, taking into account that high concentrations of these plant regulators can cause severe adverse effects on plant growth, controlling their concentration in vinasse is highly recommendable.

## Figures and Tables

**Figure 1 ijms-23-09976-f001:**
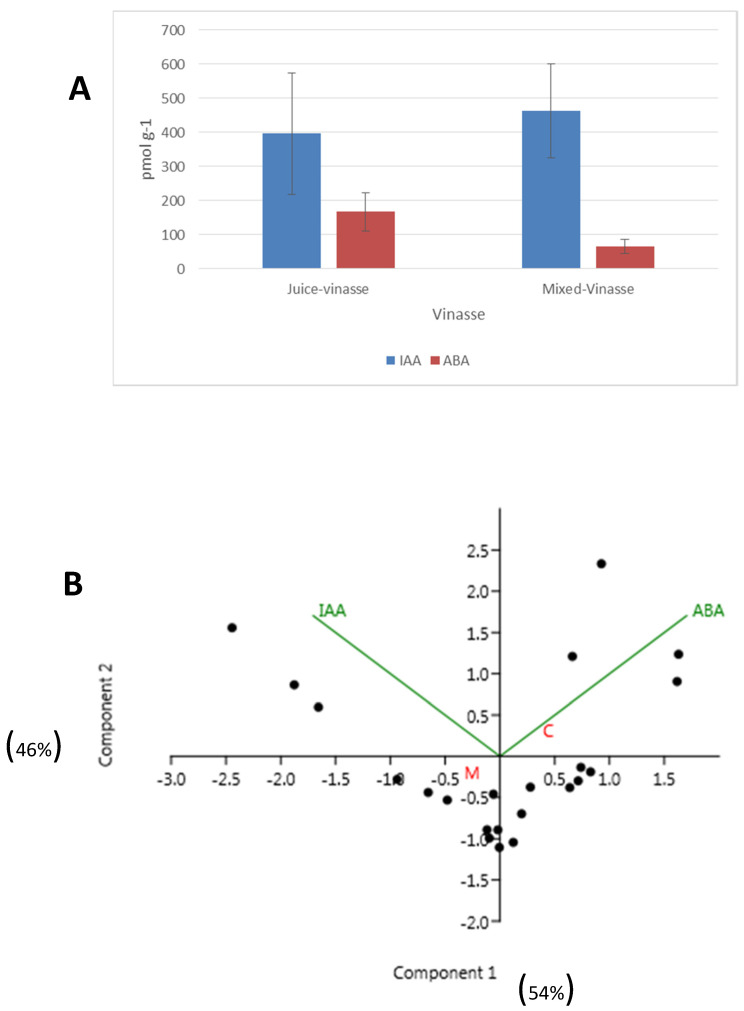
(**A**) Average concentration of IAA and ABA in different types of cane vinasse (mean ± SE); (**B**) PCA analysis. (Black dots correspond to each cane vinasse sample).

**Table 1 ijms-23-09976-t001:** Concentration of some phytoregulators in diverse types of cane vinasse. Zeatin (Z), cis-zeatin riboside (c-ZR), isopentenyladenine (iP), isopentenyladenosine (iPR), indole-3-acetic acid (IAA), and abscisic acid (ABA). (Red—higher concentrations; yellow/green—lower concentrations.).

Vinasses			Phytoregulators			
			pmol g^−1^			
Juice	IAA	ABA	Z	cZR	iP	iPR
1	82.57	138.07	6.36	NC	38.97	0.16
2	188.71	315.91	11.30	NC	30.27	NC
3	123.43	10.69	NC	NC	NC	NC
4	1407.14	17.42	NC	NC	NC	NC
5	300.00	344.89	5.94	NC	26.60	0.13
6	51.21	8.89	0.00	0.01	0.01	0.00
7	80.80	124.62	NC	NC	0.52	NC
8	936.57	378.22	4.48	0.47	2.15	0.25
**Mixed**						
1	209.43	94.70	6.39	NC	51.95	1.00
2	166.29	17.58	0.26	NC	NC	NC
3	1234.29	13.28	NC	NC	0.33	0.06
4	131.14	25.78	NC	NC	NC	NC
5	516.00	10.55	NC	NC	0.21	NC
6	671.43	0.00	NC	NC	NC	0.04
7	81.43	156.82	6.58	0.16	41.23	0.16
8	421.71	17.42	0.71	NC	1.34	0.04
9	633.14	260.89	7.99	NC	75.37	0.69
10	123.57	60.64	NC	NC	NC	NC
11	1851.43	27.91	NC	NC	NC	NC
12	28.21	24.89	NC	NC	NC	NC
13	298.71	59.03	12.33	NC	77.83	NC
14	131.57	153.88	4.70	1.34	43.45	0.27

NC. Below detection limits.

## Data Availability

All data are included in the table we publish in the article.

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
