# Peer review of "Cane Vinasses Contain Bioactive Concentrations of Auxin and Abscisic Acid in Their Composition"

_ijms, 2022, doi:10.3390/ijms23179976_

Round 1

Reviewer 1 Report

Manuscript is well written and clear.  My comments are minor.

Line 12.  If this benefit might be due to nutrients, then "biostimulant" seems the wrong word.  "nutrition" seems more appropriate.

Line 32. Suggest ending the sentence with "milling." and beginning a new sentence with "Mixed-vinasse".  Previous text discusses two options, and this "Mixed-" text comes across as a third option, which can be confusing.

Line 109.  Given storage at 15 degrees instead of rapid freezing and deep frozen storage, was there potential decomposition of the hormones during storage and transportation?  If the sites were across Brazil, presumably there could be considerable variation in transport time. 

Lines 64-65. Is it possible to propose some potential explanations for the high variability in results?  Variable transportation time, different temperatures on-site, different sugar cane cultivars, different modes of processing, different growth stages at sampling time?

Were there any lab replications of each sample? The text implies no replicate containers at each site.

Lines 122-124. Delete the heading "5. Conclusions" and the instructions.

Is the acronym "PCA" defined anywhere?  I could not find a definition, although I know its meaning.

Author Response

Reviewer 1.

Thanks a lot for all suggestions the the Rev 1 has made.

Line 12.  If this benefit might be due to nutrients, then "biostimulant" seems the wrong word.  "nutrition" seems more appropriate.

We have eliminated this word and concept from the introduction and abastract. We only mention that both IAA and ABA might have a biostimulant role in the main mechanisms responsible for the beneficial action of cane vinasses on sugar cane yields.

Line 32. Suggest ending the sentence with "milling." and beginning a new sentence with "Mixed-vinasse".  Previous text discusses two options, and this "Mixed-" text comes across as a third option, which can be confusing.

We have introduced this change.

Line 109.  Given storage at 15 degrees instead of rapid freezing and deep frozen storage, was there potential decomposition of the hormones during storage and transportation?  If the sites were across Brazil, presumably there could be considerable variation in transport time.

We agree with the Reviewer. In order to see if this methodology may affect hormone concentrations we carried out a previous study comparing the freezer sample and the liquid-stocked sample and the results were comparable if the air is eliminated from the tank and the tank is protected from light.

Lines 64-65. Is it possible to propose some potential explanations for the high variability in results?  Variable transportation time, different temperatures on-site, different sugar cane cultivars, different modes of processing, different growth stages at sampling time?

As we mention in the text of the new version of the MS, we have observed that this variability is also obtained for other main components of vinasse, such as K or organic acids. In general, some authors proposed that this variability is principally associated with the production process used in each factory.

Were there any lab replications of each sample? The text implies no replicate containers at each site.

Yes, we had only one tank from each factory. We made analytical replications for adapting the analytical method to vinasse matrix.

Lines 122-124. Delete the heading "5. Conclusions" and the instructions.

We have introduced the change.

Is the acronym "PCA" defined anywhere?  I could not find a definition, although I know its meaning.

Yes, we have defined the acronym in the new version of the MS

Reviewer 2 Report

Authors should clarify the use of the term biostimulant. It is improper to use the term biostimulants

Author Response

Reviewer 2

Thanks a lot for all suggestions the the Rev 2 has made.

Authors should clarify the use of the term biostimulant. It is improper to use the term biostimulants

We agree with the reviewer. We have modified the text of the new version of the MS in order to introduce the changes suggested by the reviewer.

Reviewer 3 Report

In general, the manuscript is very inedited, it is written in a clear and correct way and the results are highly relevant. A single issue that could be corrected by the authors concerns the title.
The title seems extensive and informs a biostimulation capacity of the studied material that was not evaluated in the present study. A reformulation could bring more clarity to the text by being more informative.

Author Response

Reviewer 3

Thanks a lot for all suggestions the the Rev 3 has made.

In general, the manuscript is very inedited, it is written in a clear and correct way and the results are highly relevant. A single issue that could be corrected by the authors concerns the title.

The title seems extensive and informs a biostimulation capacity of the studied material that was not evaluated in the present study. A reformulation could bring more clarity to the text by being more informative.

We agree with the reviewer. We have modified the title of the MS.

Round 2
